# Study of Shear-Cutting Mechanisms on Wood Veneer

**Vicky Reichel [1,2,*]**, **Werner Berlin [1,2]**, **Felix Rothe [1,2]**, **Jan Beuscher [1,2]** and **Klaus Dröder [1,2]**

[1]  Institute of Machine Tools and Production Technology, Technische Universität Braunschweig,
     38106 Braunschweig, Germany; w.berlin@tu-braunschweig.de (W.B.); f.rothe@tu-braunschweig.de (F.R.);
     j.beuscher@tu-braunschweig.de (J.B.); k.droeder@tu-braunschweig.de (K.D.)
[2]  Open Hybrid LabFactory e.V., 38440 Wolfsburg, Germany
*   Correspondence: v.reichel@tu-braunschweig.de; Tel.: +49-531-391-65047

**Abstract:** Multi-material structures made from renewable materials are increasingly being addressed in research and industry. Especially lightweight applications based on wood and polymer materials offer an important opportunity to reduce weight and $CO_2$ emissions, and thus create a sustainable economy. When establishing new material combinations, it is necessary to take economical and efficient manufacturing processes into count to enable the market entry. Therefore, the existing manufacturing processes needed to be adapted and improved in terms of the specific machining characteristic of the wood material. This study targets a combined process where a forming and shear-cutting process is also integrated in an injection-molding tool. The findings on the shear-cutting process of wood veneers are not widely investigated yet. Therefore, process and material-related dependences like cutting velocity, tool shape design, and preconditioning of wood veneers were examined. The target values cutting force and cutting-edge quality were used to describe the relations. The results showed specific damage and fiber fractions of the wood material compared to the isotropic materials (e.g., metal). In addition, low cutting forces appeared by realizing a drawing cut and high cutting speeds. A decrease in the cutting force with a higher moisture content could not be shown for the used wood types.

**Keywords:** wood veneer; shear-cutting process; cutting edge quality; injection molding

## 1. Introduction

Due to environmental changes, renewable and degradable material solutions are gaining increasing interest as lightweight solutions in industry and academia, in order to reduce emissions (e.g., $CO_2$). One highly prioritized topic in the lightweight sector is the combination of natural fibers and wood materials with polymers. When using wood in such material combinations it is usually applied as short fiber or wood flour, e.g., in wood–plastic compound (WPC). To take advantage of the mechanical behavior of the wood structure with its anisotropic characteristic, it is necessary to investigate the material combination of polymer materials with veneer lumber [1] (p. 351).

### 1.1. Wood and Polymer Materials for Lightweight Applications

Lightweight materials are used in several industries like aerospace, automotive, or construction. The application of these materials addresses the reduction of weight and therefore $CO_2$ emissions. When using bio-based or biodegradable materials, this factor is targeted even more effectively.

The typical characteristics needed for lightweight solutions are materials with a high mechanical performance and a low weight ratio. By combining materials from different material classes like, wood and polymer, the advantages of the materials can be combined. For this specific combination, wood as a fiber-reinforced material and the formability of polymer are the main advantages [1]. The

increasing usage of (thermoplastic) biopolymers shows the relevance of the topic [2,3] (pp. 200–310). Biodegradable polymers contribute to improved recyclability of parts, which is important for disposable items as well as for long-term products (especially in the automotive industry) [4]. The combination of wood veneer and polymer is commonly used for decorative trim and surface elements used in cars, yachts, or furniture [5–8].

However, the structural benefit of the wood material is addressed in parts as well. By combining thin veneer in different directions, plywood material is generated and used when stiff materials are needed. Polymer materials (e.g., adhesive, thermoplastic films, etc.) form the bond between the veneers [9] (p. 153). By combining polymer and wood veneer, the adaption of the established manufacturing processes becomes necessary.

Typical bio-based polymers are polylactic acid (PLA), polyethylene (PE), polyamide (PA), polyethylene terephthalate (PET) and are used mainly for packaging and consumer goods but also for automotive and transports [10]. The polymers made from agricultural feedstock and biomass (e.g., corn, sugarcane, forestry) tend to be more expensive than polymers from fossil primary materials [2,11] (pp. 239–244). Especially when the use of agricultural areas for food production is converted into the production of non-food biomass, increased attention is needed [10] (p. 40, 46). Therefore, manufacturing processes need to be more efficient and bio-product performances need to be more competitive than regular properties.

The wood material characteristics are highly related to the material itself due to natural growing processes. In addition, moisture has a high influence on the mechanical behavior and therefore on the forming, cutting, and bonding processes [9] (p. 75). Thermoplastic polymers as used in the injection-molding process show decreasing mechanical properties in presence of humidity [12]. Biopolymers show higher dependencies on processing parameters (e.g., temperature and demolding behavior) [13]. Resulting from these arguments the commonly used the injection-molding process is not suitable for the manufacturing of wood veneer-polymer parts and needs to be adapted.

*1.2. Process Integration*

Injection molding is a common manufacturing technology to process polymers. The process offers high quantities in short cycle times and is therefore suitable for high volume production. In Figure 1a, the concept of an injection-molding mold is shown. When the mold closes, the cutting frame is moved and it bends the veneer at the die. When the bending limit of the veneer is reached, the frame cuts the material. The polymer injection follows when the mold is closed. Therefore, the cutting process and the injection-molding process correlate directly. Depending on the cutting behavior of the wood material, specific cutting edges appear and influence the bonding between the different materials. A possible hybrid polymer–wood structure is depicted in Figure 1b. The targeted material combination consists of a cut veneer with polymer edge finish and over-molded polymer ribs acting as a structural reinforcement and functionalization opportunity (e.g., attachment points).

The concept combines the forming and shear-cutting of the veneer lumber and a subsequent polymer application. This offers the advantage to save process time, while ensuring high part quality. The shear-cutting method is well-known to cut sheet metal in a short process time and to be in high quality. When targeting a combination of all three processes (forming, injection molding, cutting) in one concept the shear-cutting process is the only possible cutting process to be integrated. Hence, it is necessary to comprehend the different behavior of wood materials in this process, compared to sheet metal.

This paper aims to investigate the open shear-cutting process of wood veneer. The fundamental findings and trends include the influences of the shear-cutting method, as well as the material-related influences of wood. This work experimentally investigated the cutting force, under the influence of cutting angles, cutting speed, wood type, and moisture content. The results are analyzed in the context of a subsequent injection process after cutting.

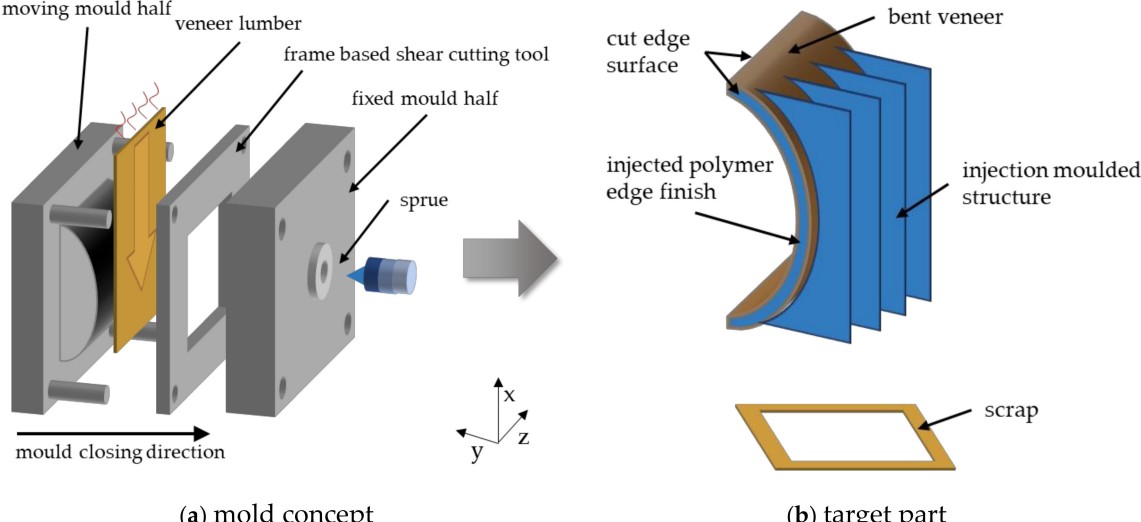

(**a**) mold concept

(**b**) target part

**Figure 1.** Injection molding mold concept (**a**) and target part characteristic (**b**).

### 1.3. Shear-Cutting Method

Shear-cutting is used for the high volume processing of two-dimensional semi-finished products. The cutting process is often combined with forming operation to enable short process cycles. One stroke of the cutting tool leads to a complete formation of the cutting edge. This allows a high amount of automated cuts to be carried out in a short time [14,15]. Often the process is combined with forming operations in one tool [16]. A disadvantage of the shear-cutting process is the low flexibility for the cutting shape. The whole cutting tool (punch and die) needs to be changed at the tool guidance to vary the shape of the parts. Since shear-cutting is a tool-based method in contrast to water-jet or laserbeam, cutting wear appears on the punch and die, such that maintenance intervals need to be considered. In addition, the cutting forces are applied on the cutting edge of the workpiece and leads to a specific cutting-edge geometry (cf. Chapter 3.1). In accordance with DIN 8588 [17], shear-cutting is a chipless cutting process.

In shear cutting, the moving punch tool passes the static die (cf. Figure 2a). The clearance between these tools is called the die clearance, which influences the cutting force, wear of the tool, and the workpiece edge cutting quality (cf. Chapter 3.1). These factors are in conflict with each other when optimizing the process. A smaller die clearance increases forces and tool wear but also improves surface qualities by lowering the roll over and the torn zone. Cutting forces decrease by widening the clearance and usage of lubricants but can also be lowered by decreasing the wedge angle $\beta$, which reduces the frictional forces [16,18,19].

Influencing the cutting force is also possible by using different punch geometries (Figure 2c). By increasing the cutting angle $\alpha_c$, a drawing cut is executed. The load applied on the cutting edge of the workpiece is more selective than that for a straight cut, where the whole cutting edge is cut at the same time. When comparing the maximum cutting forces of the two punch geometries, the force for a straight cut is therefore higher.

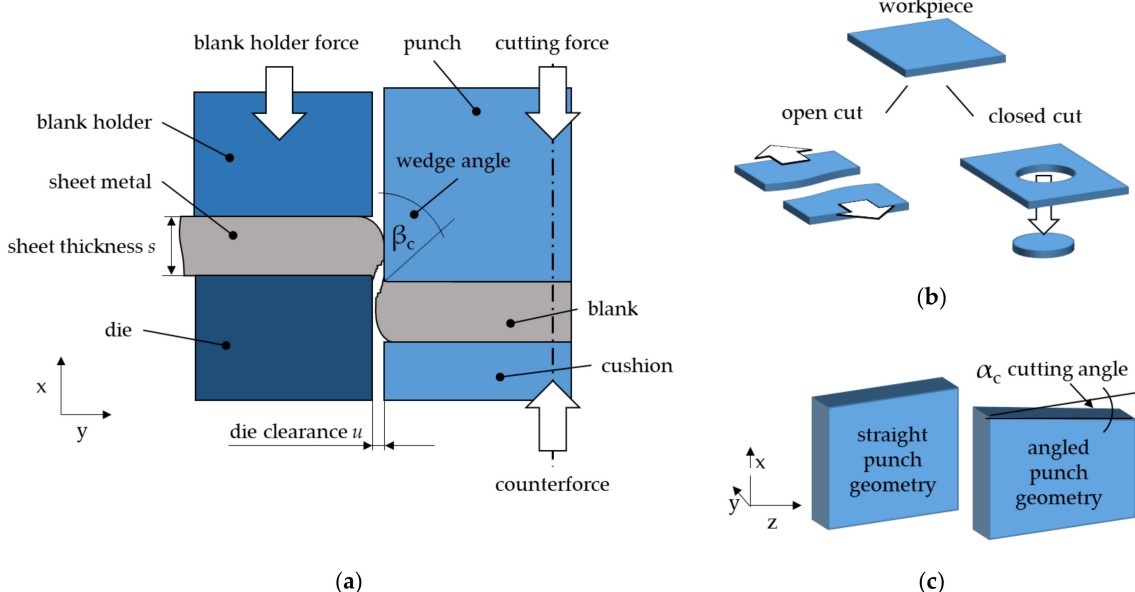

**Figure 2.** Basic terms of the shear-cutting process (**a**), and definition for open and closed cut (**b**) according to [17,20] and punch geometry (**c**).

### 1.4. Cutting Process of Wood Materials

The common separation process for wood materials is sawing [21–23]. However, this chip-forming process has different movement characteristics and therefore it cannot be compared to shear cutting. In the sawing process, material is gradually removed from the workpiece, which is no longer available in the final component.

There are minor findings on the closed shear-cutting on wood materials. Kollmann [23] listed findings for the tool geometry and cutting angle optimized for low forces and clean cutting edge of a closed cut on different wood materials. Recent findings on the closed shear cut (definition cf. Figure 2b) of medium density fiberboards (MDF) can be found in [24]. Due to the different cutting mechanisms and structure of the investigated wood materials, these results cannot be transferred to the questions targeted in this research.

A similar process to shear-cutting is knife cutting, which is used for example to cut thin veneer lumber [9,23]. The difference to shear-cutting is a smaller wedge angle of the knife tool and no die is needed [25] (p. 56). The small wedge-angle leads to a lower influence on the damage of the material [26] (p. 14). This leads to the disadvantage that the small wedge angle leads to a higher wear dependency than shear cutting.

There have been numerous studies on the cutting mechanisms of wood through common methods like sewing and milling [26–29], as well as alternative cutting methods like water-jet [30–32] and laser cutting [33–36]. The potential of the shear-cutting on wood has not been extensively investigated in literature, so far. There are no recent findings on favorable shear-cutting parameters of wood veneers, due to the uncommon application of shear-cutting or stamping on this material [9,23]. By investigating the behavior and cutting edge quality of veneer lumber in the shear-cutting process as a function of the material and process parameters, fundamental knowledge is acquired.

Defining the forces in the shear-cutting process of veneer lumber is the major question for characterizing the shear-cutting behavior of wood materials. Which cutting-edge characteristic is suitable for the subsequent injection-molding process is not known yet. Therefore, targeted properties of the cutting edge are not defined for the overall process described in Chapter 1.1. The optimization to a minimal cutting force value while considering the cutting-edge quality is the target of the experiments.

## 2. Materials and Methods

### 2.1. Specimen

The wood material used in this investigation was veneer lumber from Templin OHG, Winsen, Germany. A knife guillotine process out of a pre-sectioned trunk was used for cutting the veneers. As this material grows naturally, the resulting veneers differ in the appearance of annual rings. Therefore, anisotropic mechanical characteristics appear.

To show the influence of these characteristics, experimental studies on different grain orientations: 0°, 45°, and 90° were conducted. Additionally, the specimens were conditioned differently to show the influence of moisture relating to the wood types. The wood types used for the experiments were beech, birch, oak, and maple, with a thickness of 2 mm. The specific density of the wood types influence the cutting force [26] (p.21). As a basic material, veneer strips were conditioned and cut. The processed specimen and its dimensions are shown in Figure 3. The investigated cutting-edge length was 60 mm.

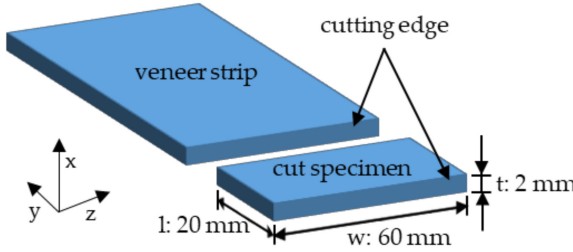

**Figure 3.** Specimen dimensions.

### 2.2. Experimental Setup

The design of experiments for shear-cutting on wood veneer was built with respect to important findings on the parameters related to a subsequent injection-molding process. The investigated parameters of this study were the following:

- Tool-related—punching tool shape.
- Process-related—cutting angle to grain orientation, cutting speed, testing temperature.
- Material-related—wood kind and moisture content.

As described in Chapter 1.2, different tool shapes led to different cutting mechanisms and cutting forces. The investigated tool shapes allow realizing a drawing and pushing cut. The different grain orientations need to be realized due to the circulating cut of the veneer sheet in the target part. The cutting speed influences the overall process time, which should be minimized [26] (p. 14). To show the dependencies of the cutting-edge quality and cutting speeds was the target of the investigation. When varying the testing temperatures, the correlation of the mold temperature enclosing the veneer was simulated. As wood is a hygroscopic material, the variation of moisture content was also included in the design of experiments. To complement this investigation, the moisture content of the veneer specimen itself was also varied. Depending on the wood type, this could have different effects on the mechanical characteristics.

### 2.2.1. Shear-Cutting Tool

The shear-cutting process was realized on a self-designed shear-cutting tool. One stroke was about 10 mm. Hence, specimens up to 9 mm thickness were reliably workable. In addition, a blank holder fixed the workpiece in its position while the punch was moving. This increased the cutting-edge quality and helped to avoid tipping of the veneer strip. Additionally, the cutting tool allows to differ the die clearance between 0.016 mm to 0.032 mm. The die clearance should take 5% to 12% of the sheet thickness [19] (p. 364). For the wood veneer used (thickness *t*: 2 mm), this led to a die clearance of

*u*: 8%, which is suitable for processing the veneer specimen. The shear-cutting tool was mounted on a standard testing machine with a load cell (up to 50 kN), which realized the movement of the tool. Since the load cell is a sensitive measuring device, it was possible to detect reliable and reproducible forces and setups.

### 2.2.2. Process Parameters

Different single layer veneer lumber specimens were cut to investigate the cutting-edge, while documenting the maximum cutting force $F_c$ in N as the target variable. All process parameters and values can be found in Table 1.

**Table 1.** Used process parameters of the experiments and variation steps.

| Process Parameter | Symbol & Units | Variation |
|---|---|---|
| Specimen dimensions | $w \times l \times t$ in mm | $60 \times 20 \times 2$ |
| Shear gap | $u$ in mm | 0.016 (u/t: 8%) |
| Cutting speed | $v_c$ in mm/min | 10; 50; 100; 200; 300; 400; 500 |
| Tool shape/cutting angle | $\alpha_c$ in ° | straight: 0; angled: 20 |
| Grain orientation | $\gamma_c$ in ° | 0; 45; 90 |
| Wood type | - | beech (BE); oak (OA); birch (BI); maple (MA) |
| Testing temperature | $T_t$ in °C | 23; 50; 80 |
| Preconditioning (according to EN 13483-1) | $\omega$ in % | 9; 15 |
| Quantity of specimen per series | - | 10 |

## 3. Results

### 3.1. Cutting-Edge Surface Characteristics of Veneer Wood in Comparison to Metal

When investigating the cutting edge of the workpiece, there were different characteristic sections measured. The relation of these sections indicated the quality of the cut. Since shear-cutting is mainly used for working metal, there are only guidelines and standards for this material. To enable the evaluation of the veneer specimen, the standard for steel material was applied (VDI 2906 [37]). Typical shear-cutting edges of both materials are displayed in Figure 4.

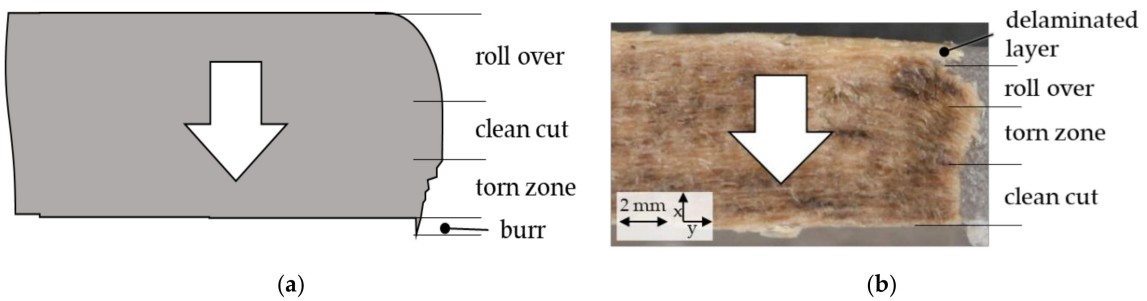

|     |     |
|:---:|:---:|
| (**a**) | (**b**) |

**Figure 4.** Cut surface characteristic values for steel (**a**) according to [25] and beech wood (thickness 5 mm; $\gamma_c$: 90°) (**b**).

In the shear-cutting process, a bending load occurs, resulting from the shear forces in different planes when shearing the workpiece. This leads to a bended surface that is called roll over. While extending the shear gap, the roll over zone enlarges. A high quality cut has a high ratio of a clean cut, due to an adverse dimensional deviation and high surface roughness in the roll over and the torn zone. The torn zone describes a section where a clear cut is not executed but the material is ripped. Therefore, an undefined surface occurs. The burr section is one of the main negative aspects in shear-cutting steel materials. The appearance of the burr influences the subsequent process steps (e.g., joining) and

often requires a cost intensive post-processing of the cutting edge (deburring). Figure 4b shows a micro section view with the cutting edge of 5 mm wood veneer shown. Comparing the veneer to the steel surface, it was observed that the torn zone where the fibers tear apart was situated behind the roll over zone where the fibers were bent. A clean cut appeared at the lower cutting section. It was assumed that the fibers were first compressed and stretched to their maximum, followed by a clear cut. This hypothesis was supported by the deviation in length that the fibers showed in the torn zone. The bended fibers had larger length deviation than the fibers in the clean-cut section, when they sprung back to their initial location. In addition, a delamination in the first millimeters could be detected. The fibers in this section sprung back when the maximum load exceeded and tearing appeared. This effect led to a delamination between the torn fibers and the expanded and bent fibers. In comparison to steel materials, the shear cut of wood did not lead to a burr.

*3.2. Surface Characteristics of Wood Veneer*

The experiments were realized with specimen thickness of 2 mm (cf. Figure 5). The specific zones could be detected clearly but were much smaller than for 5.0 mm. The different sections were measured to find relations. It emerged that the deviation on the values was too high to identify significant effects of the process parameters to these sections. Especially, the roll over and the torn zone were hard to identify. The results for the influence parameters on the cutting-edge quality are shown below.

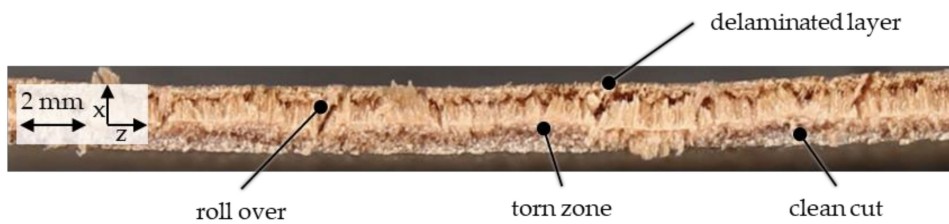

**Figure 5.** Surface characteristics for 2 mm beech specimen, $\gamma_c$: 90°.

### 3.2.1. Influence of Grain Orientation

Wood veneer has anisotropic characteristics, which are highly related to the cutting behavior [38,39]. In the cutting process, a circulating cut at all sides of the part is targeted. This necessarily leads to cut the veneer in all angles from 0° to 90° fiber orientation. Therefore, three different angles were investigated to show the quadratic coherence. The results are shown in Figure 6.

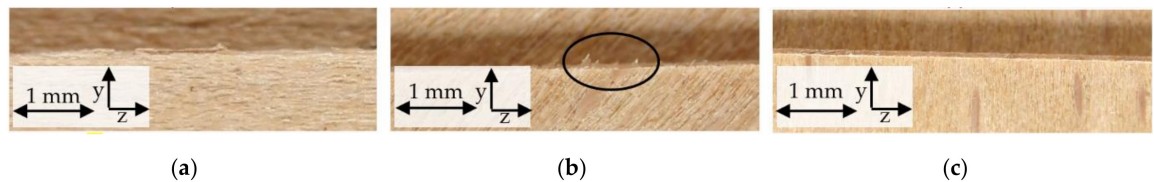

**Figure 6.** Cutting-edge behavior for different fiber cutting angles on beech veneer, top view. (**a**) $\gamma_c$: 0°; (**b**) $\gamma_c$: 45°; and (**c**) $\gamma_c$: 90°.

All three cutting edges showed acceptable results of fiber fringes on the cutting edge. Cutting in $\gamma_c$: 0° meant that the fibers were in a parallel position to the cutting-edge of the tool. For these specimens, minor dimensional deviations were detected. For the specimens with a 45° grain orientation, a slightly frayed surface was detected. This was due to the angle of the punch to the fibers, which tend to be pushed away from the initial position. By moving away, the fibers cannot be cut and get ripped. When moving back, these fibers were longer than the cut ones. The most sharp and clean surface was cut on $\gamma_c$: 90°. The fibers were fixed in their position due to the orthogonal angle they have to the moving direction of the tools.

### 3.2.2. Influence of Punching Tool Shape

The results on the investigations of the influence of the tool shape, and more precisely the cutting angle $\alpha_c$, are shown in Figure 7 for $\gamma_c$: 90°. Comparing the cutting edge, it can be seen that an angled cut led to more fiber damage (Figure 7a). This is related to the applied force per section, which was higher for the angled cut (cf. Chapter 1.2.). For a cutting angle of $\alpha_c = 0$° (straight cut), the fiber breakage and delamination appeared to be in lower intensity (cf. Figure 7b,d).

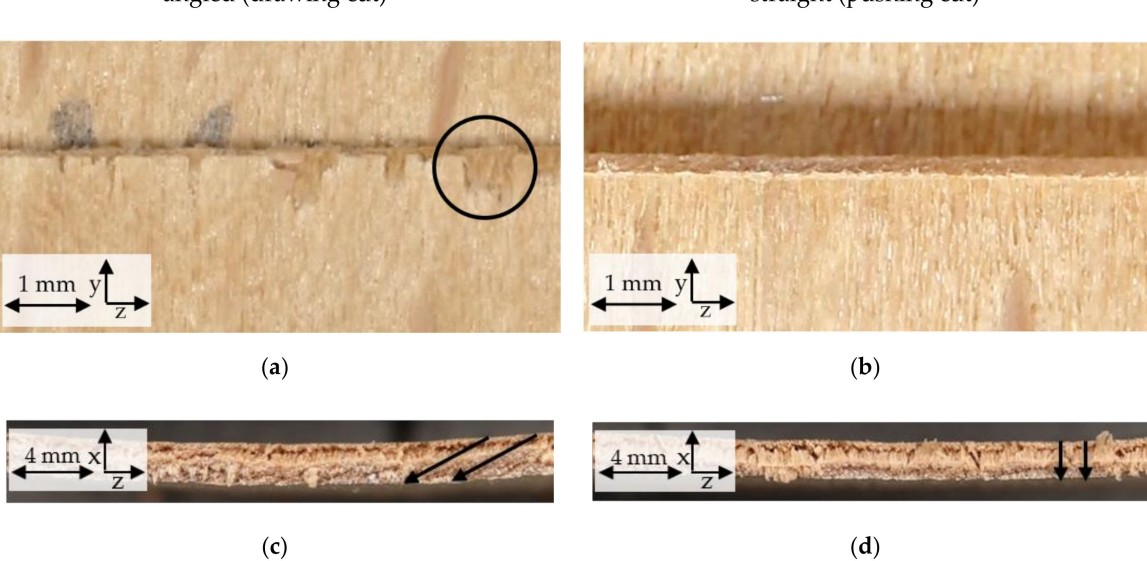

**Figure 7.** Results on the surface characteristics for the angled and straight punching tool shape, $\gamma_c$: 90°. (**a**) Top view on the cutting edge; (**b**) top view on the cutting edge; (**c**) front sectional view on cutting edge; and (**d**) front sectional view on cutting edge.

Comparing the section views (Figure 7c,d), it was observed that the torn zone was much smaller for the angled cuts. There was no force applied on the sections to compress the veneer when the tool shape was angled. Further, it was detected that the fiber direction differed due to the angled movement of the punch (cf. arrows Figure 7c,d). The results for $\gamma_c$: 0° did not show relations on the cutting angle. The top views did not change when varying the punch. Especially when comparing the sectional views, the fibers were not influenced in their orientation or the surface characteristics of the cutting edge changed. This was related to the composite structure, while the fibers in $\gamma_c$: 90° were not fixed in their longitudinal position, the fibers in $\gamma_c$: 0° direction were not sheared. The cut led to a shearing in between the fibers. Even when fibers were sheared because of the natural given deviations in the fiber orientation of a specimen, this led to fiber fractions and a frayed cutting edge but not to a difference in the surface characteristics.

### 3.2.3. Influence of the Cutting Speed

The cutting speed $v_c$ is one of the most significant parameters when optimizing the cutting process in terms of economic efficiency. In our special case of an integrated process, this meant that the velocity of the punch moving against the die simultaneously also represented the closing velocity for the injection-molding mold. It could be detected that the cutting speed had no significant influence on the surface of the cutting edge. Hence, the cutting process could be adapted to the requirements of a short process time and the integrated process.

### 3.2.4. Influence of Wood Type

The comparison of different wood types showed the relation on the density of different wood types. The specific ratio of the density of early and late wood, led to an undulating cutting edge. The low-density early wood was less stiff and showed more fiber breakage (cf. Figure 8d). Dependencies on the grain orientation could not be shown clearly. When the cutting direction was $\gamma_c$: 0° there was no reliable difference of surface characteristic for the late or early sections in the specimen. As mentioned in 3.2.2, this relates to the composite structure of wood.

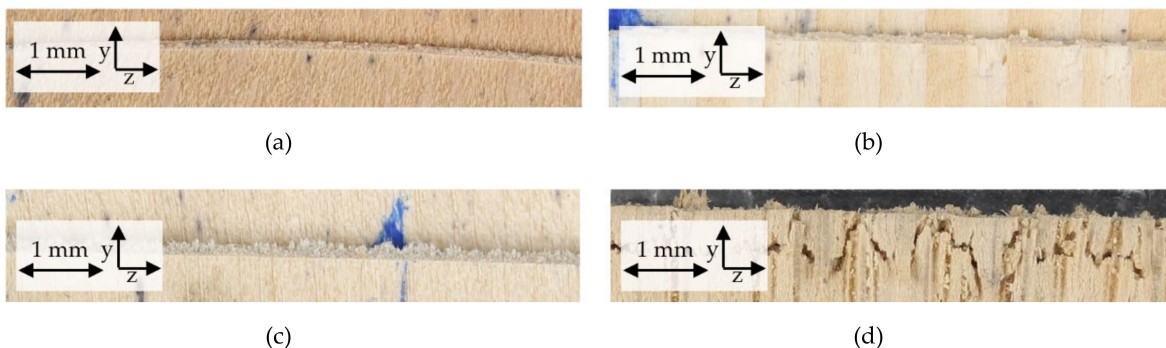

(a)                                             (b)

(c)                                             (d)

**Figure 8.** Results on the surface characteristics of different wood types, $\gamma_c$: 90°. (**a**) Beech, top view; (**b**) birch, top view; (**c**) maple, top view; and (**d**) oak, top view.

### 3.2.5. Influence of Moisture Content

When increasing the moisture of the veneers, the structure of the specimen changes. The wood structure becomes more bendable [9,40]. When comparing the cutting-edge surfaces (cf. Figure 9, $\gamma_c$: 90°), it could be observed that the torn and clean cut zones were indistinguishable. A higher ratio of fibers become bendable, which leads to a higher dimensional deviation of the overall cut, as the section at the upper cutting-edge detects more fiber breakage (Figure 9b). The investigations on the grain direction of $\gamma_c$: 0° showed no differences of surface characteristics.

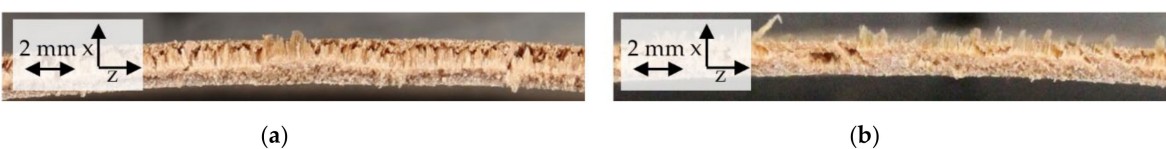

(**a**)                                             (**b**)

**Figure 9.** Surface characteristics for beech specimen at 9% (**a**) and 15% (**b**) moisture content, $\gamma_c$: 90°. (**a**) $\omega$: 9%; (**b**) $\omega$: 15%.

### 3.2.6. Influence of Testing Temperature

The specimens were tested at 23 °C; 50 °C, and 80 °C. Experiments under the testing temperature of 80 °C showed a different cutting edge (cf. Figure 10, $\gamma_c$: 90°). The roll over zone minimized while the clear-cut zone enlarged. The torn zone was no longer detectable. This could have been caused by a lower formability, due to the lack of moisture in the wood component at higher temperatures. Furthermore, there was no influence on the surface characteristic detectable for $\gamma_c$: 0°.

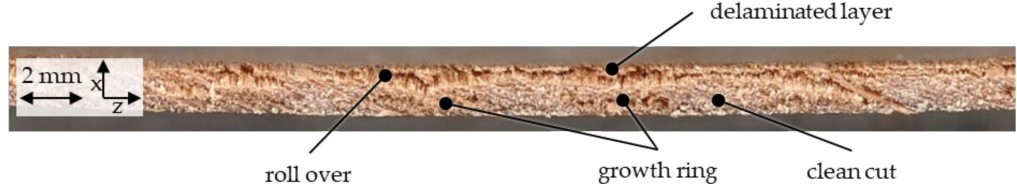

**Figure 10.** Surface characteristics for beech specimen at 80 °C testing temperature, $\gamma_c$: 90°.

### 3.3. Cutting Forces

#### 3.3.1. Influence of Punching Tool Shape

Minimizing the cutting force was an important factor due to the wear and application of a shear-cutting tool on press machines. When using different cutting tool shapes, the optimization of the cutting force was possible. It could be detected that an angled cut (pulled cut) reduced the cutting force by about 85%, in comparison to a straight cut (pressing cut). The deviation remained comparable and overall, it was low. See Figure 11 for specific process parameters and results.

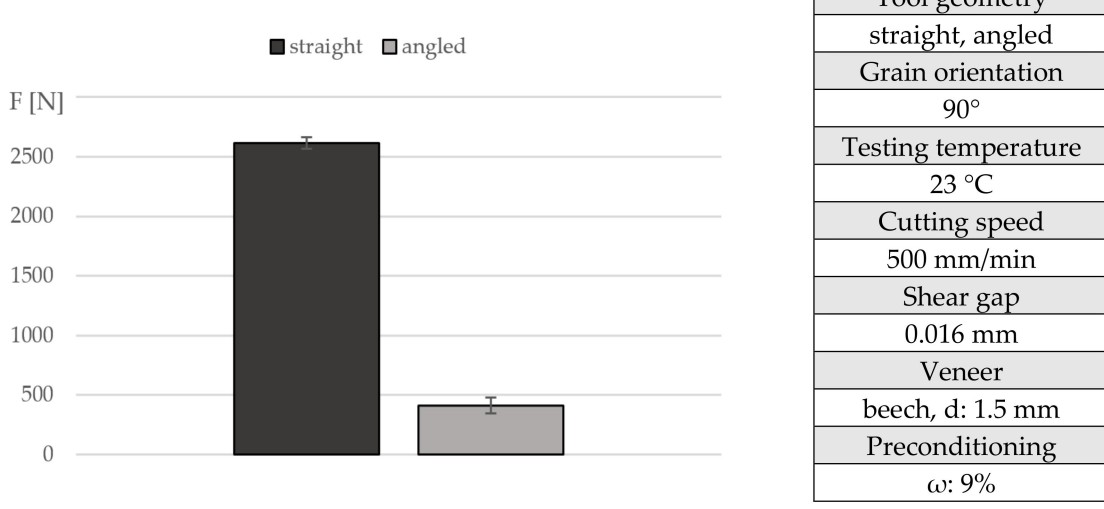

**Figure 11.** Maximum cutting force values for the punching tool shapes.

When investigating the cutting-edge damage, it could be seen that the pulled cut led to fiber tearing and an unclean cutting edge. The criteria for the cutting-edge quality were currently not defined for the target process. Due to the combination with the injection-molding processes, it is currently not known how the characteristic of the cutting-edge surface affect the bond between the wood component and the polymer. It could be assumed that an increased bonding surface area improved the bonding strength.

#### 3.3.2. Influence of Grain Orientation and Growth Rings

The highest cutting force could be detected at $\gamma_c = 90°$, when compared to $\gamma_c = 0°$ for all kinds of wood at all process parameters. This was caused by the cellular structure of the wood (tracheids, growth rings) [9] (p. 107, 126). In Figure 12, the values for beech veneer are shown. As expected, the grain orientation and therewith the amount of dense late wood growth rings influenced the cutting force. The standard deviation for $\gamma_c = 45°$ values was higher than that for the other values, which could be referred to the specimen itself. This specific grain orientation led to a different amount of growth rings and therefore to varying strength behavior.

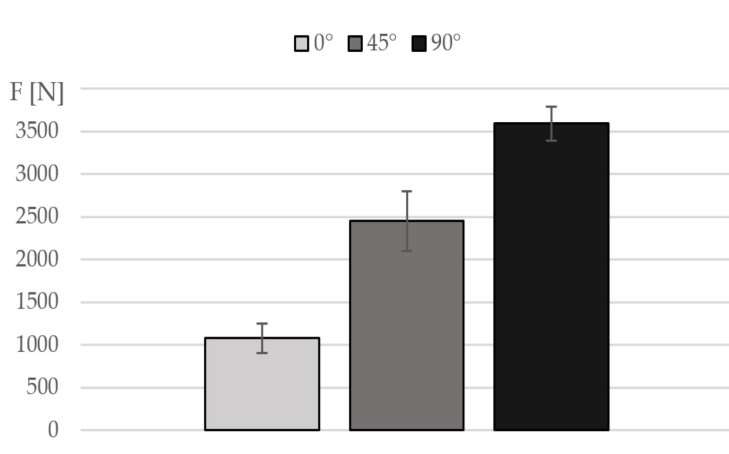

| Tool geometry |
| :---: |
| straight |
| Grain orientation |
| 0°; 45°; 90° |
| Testing temperature |
| 23 °C |
| Cutting speed |
| 500 mm/min |
| Shear gap |
| 0.016 mm |
| Veneer |
| beech, d: 1.5 mm |
| Preconditioning |
| ω: 9% |

**Figure 12.** Maximum cutting force values for grain orientation.

Due to the correlation between the number of growth rings and the cutting force, investigations comparing the values for 10 ± 2 growth rings and 20 ± 2 growth rings were carried out. The results are shown in Figure 13. It could be detected that by halving the amount of growth rings to around 10, the cutting forces were 23.86% lower.

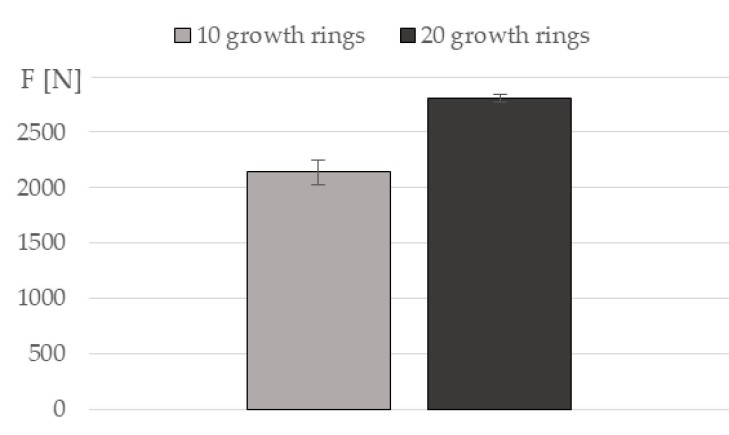

| Tool geometry |
| :---: |
| straight |
| Grain orientation |
| 90° |
| Testing temperature |
| 23°C |
| Cutting speed |
| 500 mm/min |
| Shear gap |
| 0.016 mm |
| Veneer |
| beech, d: 1.5 mm |
| Preconditioning |
| ω: 9% |

**Figure 13.** Maximum cutting force values for growth rings.

Consequently, investigations on the influence of the temperature on the number of growth rings were made additionally. A trend of the values for decreasing the influence of growth rings with higher temperatures was determined. The test on significance showed that the values were not reliable. Further investigations are needed to show what influence the temperature has on growth rings and the early and late wood sections.

### 3.3.3. Influence of Cutting Speed $v_c$

The influence of cutting speed was examined for $v_c$ = 10 mm/min to 200 mm/min. The results shown in Figure 14 indicate a regressive trend of the cutting force for higher cutting speeds. Additionally, it was also investigated if the deviation of values lowered higher cutting speeds. There was no significant effect on this between the values of $v_c$ = 300 mm/min to $v_c$ = 500 mm/min. The cutting-edge quality

was increased as well with the higher cutting forces. This phenomenon was related to the properties of the wood structure. When shearing materials, the cutting resistance relate to the tensile strength properties. The strength values depend on the testing velocity, especially for wood materials [41].

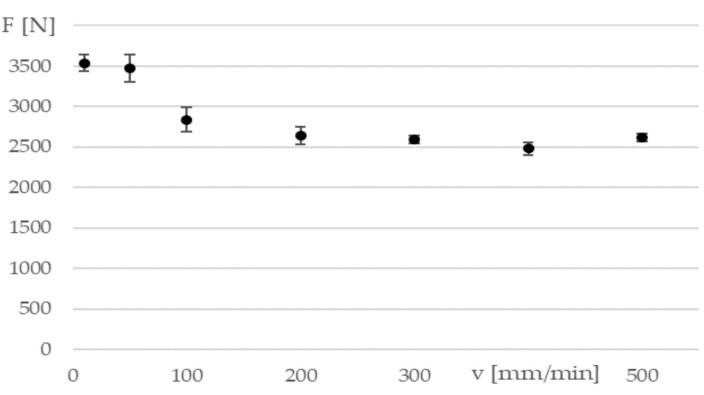

| Tool geometry |
| :---: |
| straight |
| Grain orientation |
| 90° |
| Testing temperature |
| 23 °C |
| Cutting speed |
| 10…500 mm/min |
| Shear gap |
| 0.016 mm |
| Veneer |
| beech, d: 1.5 mm |
| Preconditioning |
| ω: 9% |

**Figure 14.** Maximum cutting force values for the different cutting speeds.

The experiments showed that increasing the cutting speeds to more than $v_c = 300$ mm/min did not have a significant effect on the cutting force. Due to the process efficiency, it was expedient to reduce the process time by maximizing the speed.

### 3.3.4. Influence of Wood Type

It is known that different kinds of wood show different processing behavior due to their specific growing characteristic and structure [28]. In the investigations, only deciduous wood was tested. Figure 15 shows that birch induced the lowest cutting force $F_c$ at $\gamma_c = 90°$, while maple showed the highest value. For all kinds, there was a significant effect on $F_c$ between $\gamma_c = 90°$ and $\gamma_c = 0°$ but the maple specimens showed the highest reduction on Fc from 90° to 0° grain orientation, at about −79.54%. The highest standard deviations were detected for maple veneers as well.

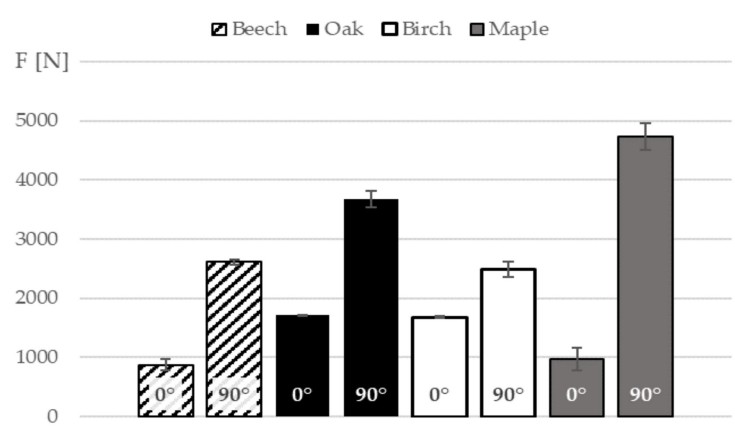

| Tool geometry |
| :---: |
| straight |
| Grain orientation |
| 0°, 90° |
| Testing temperature |
| 23 °C |
| Cutting speed |
| 10…500 mm/min |
| Shear gap |
| 0.016 mm |
| Veneer |
| beech, oak, birch, maple |
| Preconditioning |
| ω: 9% |

**Figure 15.** Maximum cutting force values for wood types and grain orientations.

For the design and integration of the cutting process, it is important to minimize the cutting forces and to keep the force constant over the entire cutting edge. In the experiments conducted, beech veneer meets these requirements the best.

### 3.3.5. Influence of Moisture Content

Figure 16 shows the influence of moisture content on the cutting force. The specimens of $\gamma_c = 90°$ wet birch and maple show an upward trend of the cutting force but with regards to standard deviation, the effect is not significant. However, the cutting force on $\gamma_c = 90°$ of wet beech (–34%) and oak (–23%) lowered more in comparison to birch and maple. This could be referred to the high moisture content of beech and oak (<90%). All specimens showed a decrease of the cutting force on $\gamma_c = 0°$. The most significant decrease for $\gamma_c = 0°$ could be detected on birch veneer (–75%). The highest declination was explored for wet maple veneer from 90° to 0° = –85.89%.

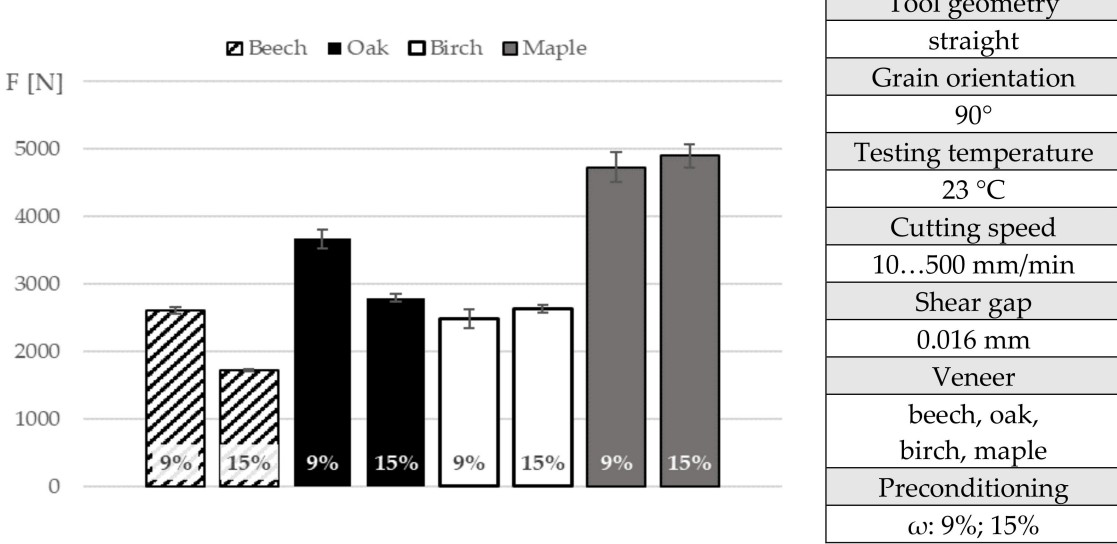

**Figure 16.** Maximum cutting force values for moisturized wood types.

Regarding the moisture influence, it was shown that for beech veneer and oak veneer, the cutting force decreased (beech: –34%; oak: –24%) with higher moisture content. Another relation was detected for the birch and maple veneer, which showed an increasing cutting force (birch: +3%; maple: +6%). The deviations showed that only the values for beech and oak were significant.

There have also been investigations on the cutting force for grain orientation (0°; 90°) and the influence of moisture content. It could be determined that all veneers showed the same behavior as a result of decreased cutting forces, between 81% (oak) and 85% (maple), in comparison to their initial moisture content. A regression analysis (reliability 97%) was performed to determine the significance of the measured values. In comparison, the orientation had the highest influence followed by the wood species and moisture content.

In summary, an increasing moisture content led to a reduction of the cutting forces. The tensile strength lowered with increasing moisture and therefore led to a reduction of the maximum cutting force [40]. The effect investigated was not reliably shown for the birch and maple veneers. Beech and oak wood have comparable mechanical characteristics, while birch and maple differ from these. The influence of moisture on the mechanical properties was related specifically to the wood type. For the overall process development, this meant the reduction of the maximum closing force of the tool. In contrast to this, the bonding of wood and polymer was negatively influenced by moisture. Specifically, thermoplastics were hydroscopic as well but showed decreasing mechanical properties in presence of water. However, in the heated injection-molding process, water molecules lead to air

pockets and might cause water vapor, which cannot escape from the closed mold. The result was a loss of precision and quality in components. In addition to this, it was assumed that high moisture levels would result in an obstacle of bonding between the wood-based material and the polymer.

## 4. Discussion

The results achieved extend the knowledge of the shear-cutting process on wood veneer. There were no specific findings available before this study. Compared to previous studies, the results showed a detailed cutting-edge surface and material damage in the working area of the cut. Since wood material is comparable to reinforced materials, some of the findings on the cutting edge might apply for e.g., fiber-reinforced plastics. The examination of the shear-cutting process on wood material showed significant differences of the surface and cutting-edge behavior, compared to metal materials. Due to its hollow and fibrous structure, wood showed more brittle cutting characteristic than metal. There was no burr zone detectable on the backside of the workpiece. This was caused by the characteristic of the fibers—there was no plastic deformation possible. The forces applied on the cutting edge were guided by the whole composite structure, so the resulting cutting edge showed approximately no interlaminar damages.

Different punching tool shapes were used to evaluate the influence of the drawing and pushing cutting mechanism. The determination of the surface edge showed that a drawing cut led to more fiber breakage but lower cutting forces, while the pushing cut showed a clean cutting-edge but higher cutting forces. This was correlated to the cutting mechanism itself. The longer the cutting length, the less force was applied on single fibers. Small cutting sections could not take the resulting cutting load and led to fiber breakage.

The influence of the grain orientation was examined by cutting in different fiber directions. When the cutting force was applied orthogonal to the fibers ($\gamma_c$: 90°), the resulting cutting force was high. Fibers are the reason for the anisotropic characteristic of the wood material and cannot take loads longitudinal to the fiber orientation ($\gamma_c$: 0°). The results showed that for the grain orientation of $\gamma_c$: 0°, the surface characteristics (sectional ratio and order) were not reliably influenced. The cutting forces correlated directly on the closing forces of the mold. The specific force per distance could be used to determine the closing force for a specific mold size. In addition, the varying cutting forces, depending on the workpiece grain orientation, could lead to a tilt motion of the cutting tool and different wear effects. The number of growth rings was a significant influence parameter for the process forces and needs to be regarded in further investigations.

Investigating the cutting speed showed a reduction of the cutting forces at high speeds. When cutting at higher speeds, the influence of the dynamic proportion of the cutting force lowered. In contrast to this, the static proportion of the cutting force correlated to the workpiece material and its shear strength. The variation of cutting speeds in the low value range was not significant to the quality of the cutting edge. When using higher cutting speeds, this might change due to force transfer between the fibers.

The different wood types showed different cutting-edge characteristics, due to their specific structure. Additionally, the appearance of early and late wood had an influence on the cutting force due to their different strength values. Using more homogenous wood types like beech and birch showed less difference in cutting forces for different cutting angles. Specifically maple showed high differences in these values and was therefore not suitable for the preferably constant cutting forces in circular cuts.

The findings on the influence of moisture were not clearly related to mechanisms in the wood material. While the beech and oak specimen showed a decreasing cutting force with increasing moisture, birch and maple materials showed the opposite. The values showed no correlation to the mechanical characteristics of the wood type. The specimen size influenced the water absorption of the specific material. When the moisture content was increased, the specimen were weighed, the dispersion within the specimen was not evaluated, and it could have had an influence. Especially the experiments on the influence of moisture were complex to perform due to the chronological order of the process

steps. Therefore, more experiments need to be done to evaluate the influence of humidity in wood on the cutting behavior. As one of the most important factors on the properties of wood and polymer, humidity needs to be described intensively. The moisture content of wood is a complex influence. By targeting the combination of an injection-molding process, this parameter should be investigated further. For this, the exact process conditions should be known (e.g., temperatures of mold and melted polymer and moisture content for specific bending ratio of the veneer).

Another important future experiment would be the determination of the bonding strength of the wood and the polymer component. The effects of the bonding mechanisms on the surface were partly investigated, but even more, the influence of the surface structure of the wood was considered to be relevant. The formation of undercuts at the cutting edge could lead to enlarged bonding areas and not only chemical but positive-fit connection. By investigating the bonding behavior of different shear-cutting surfaces to the polymer material, it would be possible to define the target structure.

The experiments did not include investigations on coniferous wood. The deformation behavior of the wood types differed due to the different strength properties (shear strength, tensile strength). In order to illustrate dependencies and interdependencies, and especially the difference to deciduous woods, these values will be determined in future experiments.

Future studies using high quality and comparable raw material would be beneficial. When presenting the results, it was shown that there were some values with higher standard deviations. Due to the standard deviation of the values, it was still appropriate to use full-factorized Design of experiments (DOE) with a high amount (≥10) of specimen per variation. It is helpful to characterize the veneer basically (density, tensile strength properties under the influence of moisture) to find dependencies. Depending on the origin, veneer manufacturing (e.g., drying process, cut) and quality of the wood used for the investigations, the relations described and the force values could differ when repeating the same experiments. When collating the current findings on wood cutting mechanisms, it was found that surface roughness was often investigated. To make the results more comparable to other cutting techniques, it would be helpful to collect surface roughness data when shear cutting.

## 5. Conclusions

The aim of this study was to investigate the shear-cutting properties of wood veneer. Regarding the parameters of the shear-cutting process (cutting angle to grain orientation, cutting speed, and testing temperature), as well as the tool-related parameters (punching tool shape) and material-related parameters (wood kind, moisture content), the cutting-edge surface and the maximum cutting force were examined. These investigations were conducted with respect to important findings for our research, aiming at the process combination of shear-cutting and injection molding. Due to the lack of knowledge about shear-cutting wood materials, the investigation provided the fundamental results of the parameter correlations, the process forces, and the cutting-edge qualities.

Analyzing the cutting edge of the specimens showed that the wood materials behave differently at every single specimen, due to its natural origin. However, correlations of the parameters and the main characteristics could be shown. When shear-cutting wood, the material showed a specific cutting-edge surface characteristic. The micro-sectional analysis showed that the different sections varied or disappeared when different process, tool, or material parameters were used. The first millimeters were highly influenced by the applied forces of tension loads. These loads led to fiber breakage, especially when a drawing cut with an angled tool was used. The roll over zone was where the bended fibers could be seen without breakage. The torn zone in which the fibers bent, compressed, and cut the deviation of the cutting edge, was the highest. Following these sections, a clear cut-zone was detected. The different sections changed their dimensions when different parameters were used.

The findings were used to analyze the conflict of low cutting forces and high cutting-edge quality, as is usual for the shear-cutting process. The results in Table 2 show the high potential of the conflict between the parameters in a shear-cutting tool. Low cutting forces appear at higher cutting speeds with an angled tool shape. Additionally, a higher moisture content led to low cutting forces. When

looking one step further and including the targeted injection-molding process, the moisture content was the most difficult variable due to the bonding of wet wood and polymer. In the investigations, beech showed the lowest cutting forces. For the cutting-edge quality, a straight cut with a minimum of tear or broken fibers was targeted. This appeared for cutting orthogonal to the fiber direction and with a straight shaped punch.

**Table 2.** Investigation results for the cutting forces and the cutting-edge quality.

|  | Low Cutting Forces | Highest Cutting Edge Quality |
|---|---|---|
| Cutting Speed $v_c$ | + | + |
| Cutting Angle $\alpha_c$ | 0° | 90° |
| Tool Shape | angled | straight |
| Moisture Content | wet | not defined |
| Shear Clearance | + | − |
| Wood Type | beech | maple |

**Author Contributions:** Conceptualization, V.R. and W.B.; Formal analysis, W.B.; Investigation, V.R.; Methodology, V.R.; Resources, J.B.; Supervision, J.B.; Validation, V.R. and F.R.; Visualization, V.R.; Writing—original draft, V.R., W.B. and F.R.; Writing—review & editing, J.B. and K.D. All authors have read and agreed to the published version of the manuscript.

**Funding:** This research received no external funding.

**Acknowledgments:** We acknowledge support from the German Research Foundation and the Open Access Publication Funds of the Technische Universität Braunschweig.

**Conflicts of Interest:** The authors declare no conflict of interest.

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
