# Peer review of "Study of Shear-Cutting Mechanisms on Wood Veneer"

_forests, doi:10.3390/f11060703_

Round 1

Reviewer 1 Report

The submitted manuscript presents an interesting study, however major revisions are needed before the manuscript could be considered for publication.

1/ The most important comment is that there is no scientific discussion in the article; the authors limit themselves in chapter 3 only to the description of the achieved results. If a discussion appears in chapter 3, then only the generally known fact and not the achieved results are discussed with the other authors.

You can find requirements for the discussion directly in the Instructions for Authors of the journal Forests:  “Authors should discuss the results and how they can be interpreted in perspective of previous studies and of the working hypotheses. The findings and their implications should be discussed in the broadest context possible and limitations of the work highlighted. Future research directions may also be mentioned. This section may be combined with Results.”

2/ As is mentioned in the Abstract, multi-material structures made from renewable materials are increasingly being addressed in research and industry. Especially lightweight applications based on wood and polymer materials offer an important opportunity to reduce weight, CO2 emissions and thus sustainable economy. However, no literature overview of renewable lightweight materials is is presented in the manuscript. Please enhance the introduction with overview of these materials, as is i.e. presented in:

Hýsek, Š.; Frydrych, M.; Herclík, M.; Fridrichová, L.; Louda, P.; Knížek, R.; Le Van, S.; Le Chi, H. Permeable Water-Resistant Heat Insulation Panel Based on Recycled Materials and Its Physical and Mechanical Properties. Molecules 2019, 24, 3300.

3/ Methodology: The used veneers has to me described more deeply in the methodology part (i.e. manufacturing parameters, manufacturer).

Author Response

The authors thank you for your review and the helpful advices given. Please see attachemnt for detailed response.

Kind regards, Vicky Reichel

Reviewer 2 Report

Dear Authors,

Authors deal with interesting topic suitable for the journal. But I did not see clear linkage between the research results on the properties of shear cutting of the veneer and injection moulding of the polymers. All materials and methods and also results are about quality of cutting edge and process properties of shear cutting. I suggest to change manuscript to focus only on quality of cutting edge of wood veneers and process properties of shear cutting. If would like to include also injection moulding and polymers into manuscript, also results of the of the quality of the bonding between wood an polymer should be shown. And also cutting properties and quality in the section between wood and polymer should be presented.

Abstract: In abstract should be presented also some results of the research.

Introduction:

Literature overview should be changed according to above suggestion.

Line 59.  reference should be 4 not 14.  

Material and methods

Suitable methods are used for testing quality of the quality of cutting edge and process properties of shear cutting.

Line 137: “is therefore is process is being investigated.« should be deleted.

Results and discussion

When the influence of different parameters are presented the grain orientation should also be shown (All section 3.2.).  The grain orientation should be mentioned also in all figures captions (figures 5-10).

Description parameters on figures 11 -16. Testing temperature RT – what does it mean? Room temperature or 23 °C please specify.  

Author Response

The authors thank you for your review and the helpful advices given. Please see attachment for detailed response.

Kind regards, Vicky Reichel

Round 2

Reviewer 1 Report

After carefully performed revisions, the manuscript can be submitted for publication.

Reviewer 2 Report

The manuscript is now significantly improved, so I suggest to accept for publication.